# Key residues in the VDAC2-BAK complex can be targeted to modulate apoptosis

Zheng Yuan[1,2], Mark F. van Delft[1,2], Mark Xiang Li[3], Fransisca Sumardy[1], Brian J. Smith[4], David C. S. Huang[1,2], Guillaume Lessene[1,2,5], Yelena Khakam[1], Ruitao Jin[4,6], Sitong He[4], Nicholas A. Smith[4], Richard W. Birkinshaw[1,2]*, Peter E. Czabotar[1,2]*, Grant Dewson[1,2]*

1 Walter and Eliza Hall Institute of Medical Research, Parkville, Melbourne, Australia, 2 Department of Medical Biology, University of Melbourne, Parkville, Melbourne, Australia, 3 Peter MacCallum Cancer Centre, Parkville, Melbourne, Australia, 4 La Trobe Institute for Molecular Science, La Trobe University, Melbourne, Australia, 5 Department of Biochemistry and Pharmacology, University of Melbourne, Parkville, Melbourne, Australia, 6 Research School of Biology, Australian National University, Canberra, Australia

* birkinshaw.r@wehi.edu.au (RWB); czabotar@wehi.edu.au (PEC); dewson@wehi.edu.au (GD)

**Data Availability Statement:** All relevant data are within the paper and its Supporting Information files.

## Abstract

BAK and BAX execute intrinsic apoptosis by permeabilising the mitochondrial outer membrane. Their activity is regulated through interactions with pro-survival BCL-2 family proteins and with non-BCL-2 proteins including the mitochondrial channel protein VDAC2. VDAC2 is important for bringing both BAK and BAX to mitochondria where they execute their apoptotic function. Despite this important function in apoptosis, while interactions with pro-survival family members are well characterised and have culminated in the development of drugs that target these interfaces to induce cancer cell apoptosis, the interaction between BAK and VDAC2 remains largely undefined. Deep scanning mutagenesis coupled with cysteine linkage identified key residues in the interaction between BAK and VDAC2. Obstructive labelling of specific residues in the BH3 domain or hydrophobic groove of BAK disrupted this interaction. Conversely, mutating specific residues in a cytosol-exposed region of VDAC2 stabilised the interaction with BAK and inhibited BAK apoptotic activity. Thus, this VDAC2–BAK interaction site can potentially be targeted to either inhibit BAK-mediated apoptosis in scenarios where excessive apoptosis contributes to disease or to promote BAK-mediated apoptosis for cancer therapy.

## Introduction

BAK and BAX are crucial effectors of intrinsic apoptosis, and their activation leads to mitochondrial outer membrane permeabilisation (MOMP) and apoptotic cell death [1,2]. Hence, understanding how BAX and BAK apoptotic activity is regulated is key for identifying opportunities to seize control of apoptosis. The focus of previous efforts has been primarily directed at their interactions with other BCL-2 family members [3]. However, non-BCL-2 family proteins are also emerging as important players in regulating BAX and BAK apoptotic function, in particular, the voltage-dependent anion channel 2 (VDAC2) [4].

**Funding:** National Health and Medical Research Council 1083077 (GL, MFvD), 2001406 (RWB, PEC), 2990062 (PEC), 2016894 (RWB), 1117089 (GL), 1156024 (DCSH), Bodhi Education Fund (GD), Veski (RWB), Australian Government Independent Research Institute Infrastructure Support Scheme 9000587 (all authors), Victorian State Government Operational Infrastructure Support, Australia (all authors). The funders had no role in study design, data collection and analysis, decision to publish, or preparation of the manuscript.

**Competing interests:** The authors have declared that no competing interests exist.

**Abbreviations:** BN-PAGE, blue native polyacrylamide gel electrophoresis; DMEM, Dulbecco's Modified Eagle Medium; DMS, deep mutational scanning; ECL, enhanced chemiluminescence; FBS, fetal bovine serum; GFP, green fluorescent protein; MEF, mouse embryonic fibroblast; MELB, modified egg lysis buffer; MOM, mitochondrial outer membrane; MOMP, mitochondrial outer membrane permeabilization; PI, propidium iodide; PVDF, polyvinylidene fluoride; VDAC2, voltage-dependent anion channel 2.

VDAC2 is one of 3 isoforms of the VDAC protein family that are present on the mitochondrial outer membrane (MOM) [5,6]. VDAC family proteins share the responsibility of transporting metabolites and ions across the MOM [7,8], but VDAC2 has drawn additional interest because of its role in apoptosis [4]. VDAC2 inhibits BAK-mediated apoptosis by sequestering BAK in its inactive form [4]. Hence, cells deficient in VDAC2 are hypersensitive to apoptotic stimuli [4]. This hypersensitivity is despite BAK being unable to localise to mitochondria effectively and that the mitochondria-associated pool of BAK alone cannot promote MOMP efficiently in VDAC2-deficient cells [9,10]. VDAC2 thus plays a role both in recruiting BAK to mitochondria and in restraining its activity once at mitochondria. At steady state, BAK is constitutively located on the MOM and assembles into a BAK–VDAC2 complex [11]. It has been suggested that the C-terminal transmembrane domain and the globular regions (α-helices 2 to 5) are involved in its interaction with VDAC2 [4,12]. However, the molecular details of this interaction remain unclear.

In this study, we use deep mutational scanning and chemical labelling experiments to identify key residues in the interface between VDAC2 and BAK. We establish the role these mutations have in stabilising/disrupting the BAK–VDAC2 and their consequences for apoptotic progression. We report that mutations of VDAC2 can stabilise the BAK–VDAC2 interaction and thereby impair BAK activation, rendering cells resistant to BH3 mimetic-induced apoptosis.

## Results

### Deep mutational scanning identifies VDAC2 β10–11 loop residues that interface with BAK

BAK is normally localised to the MOM where, prior to activation, it interacts with VDAC2 [4,11]. Genetic deletion of *Vdac2* impairs the ability of BAK to target mitochondria leading to a reduction in BAK cellular abundance [10,11]. VDAC2 is an integral membrane protein with a β-barrel architecture [13–15], previous work has identified β-strands 7–11 are involved in interactions with BAK [16]. To further narrow down the region on VDAC2 responsible for BAK targeting, deep mutational scanning (DMS) was performed on the cytosol-exposed region of VDAC2 β10–11 (Fig 1A).

First, we constructed a retroviral library of FLAG-tagged human VDAC2 single amino acid substitution variants spanning the residues Q165 to F180 and evaluated how these substitutions impacted VDAC2 expression level (S1A and S1B Fig). Impaired expression was observed for truncated variants, several variants with proline substitutions, and variants where membrane-facing hydrophobic amino acids were substituted for polar or charged residues (e.g., positions M166, F168, and F180) (S1C Fig). Most other substitutions appeared to be well tolerated as measured by VDAC2 expression level.

Next, we expressed BAK with an N-terminal green fluorescent protein (GFP) in VDAC2-deficient cells. In the absence of VDAC2, the GFP signal was low, but upon co-expressing wild-type VDAC2 the GFP signal was markedly elevated as GFP-BAK was recruited to and stabilised at the MOM (S1D Fig). Thus, we introduced the VDAC2 β10–11 DMS library to these cells and used GFP fluorescence as a measure for BAK recruitment and/or stabilisation at the MOM (S1E Fig). This DMS approach highlighted residues that either positively or negatively influenced GFP-BAK levels. The most pronounced effects were observed for substitutions of Ala171 (Fig 1B and 1C). In some cases, such as mutation to the midsized hydrophobic residues valine, isoleucine, and leucine, GFP-BAK levels increased, suggesting that these substitutions stabilised the BAK–VDAC2 interaction. Conversely, mutation to some larger residues, such as tryptophan and arginine, resulted in reduced GFP-BAK levels, indicative of destabilisation.

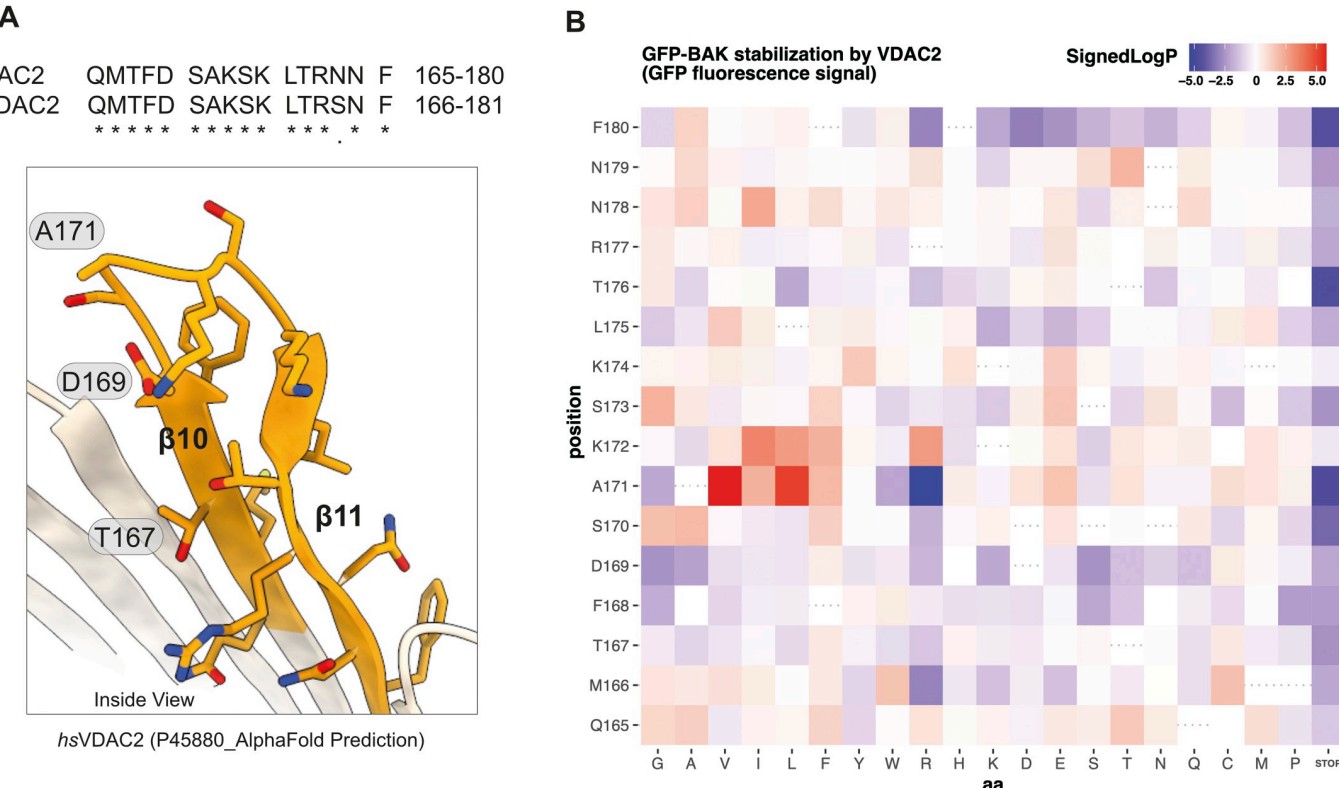

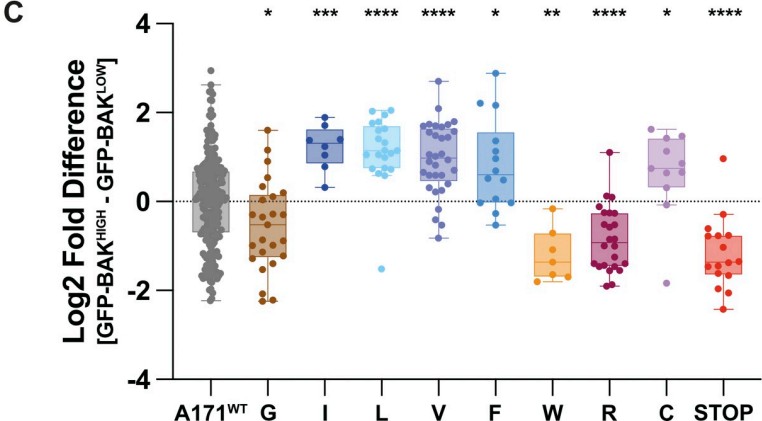

**Fig 1. DMS of VDAC2 identifies VDAC2 mutations stabilising BAK. (A)** AlphaFold2 predicted structure of human VDAC2 (Uniprot entry: P45880) [13,14] with residues within the β strands 10–11 region evaluated with DMS coloured in orange. Human and mouse VDAC2 show high sequence identity in this region but have 1 amino acid numbering difference. T167 and D169 (D168 and T170 in mouse VDAC2) were previously identified as residues for its interaction with BAK [16]. **(B)** Heatmap representation of DMS screen to identify residues within hsVDAC2 β10–11 that influence BAK recruitment and stabilisation at the MOM. $Bax^{-/-}Bak^{-/-}Vdac2^{-/-}Mcl1^{-/-}$ MEF cells were engineered with retroviruses to express GFP-tagged mouse BAK and a library of uniquely barcoded FLAG-tagged human VDAC2 variants with amino acid substitutions between positions Q165 and F180. Cells were sorted into GFP$^{low}$ and GFP$^{high}$ populations and the log2-fold difference in distribution for each barcode was determined by Illumina sequencing. Mann–Whitney $p$-values were calculated comparing the log2-fold differences of barcodes associated with each coding substitution to those associated with wild-type VDAC2 coding sequence. Data are represented as signed log transformed $p$-values to reflect the direction of change: negative/blue for variants skewed towards the GFP$^{low}$ fraction (i.e., impaired stabilisation of GFP-BAK); positive/red for variants skewed towards the GFP$^{high}$ fraction (i.e., enhanced stabilisation of GFP-BAK). **(C)** Additional detail on the site-specific mutational tolerance of VDAC2 at position A171. Each dot represents a uniquely barcoded independent clone within the DMS library and values are log2[GFP$^{high}$ barcode reads]–log2[GFP$^{low}$ barcode reads]. Aliphatic substitutions (Ile, Leu, and Val) have positive effects on BAK stability (skewed to GFP$^{high}$ fraction), while substitutions to smaller (Gly) or bulkier residues (Trp, Arg) appear to destabilise BAK in cells (skewed to GFP$^{low}$ fraction). Mann–Whitney $p$-values reflect the comparison of each substitution variant to barcodes associated with wild-type VDAC2 coding sequence ($p$: $< 0.05^*$; $< 0.01^{**}$; $< 0.001^{***}$; $< 0.0001^{****}$). The data underlying this figure is available in S1 Data. DMS, deep mutational scanning; GFP, green fluorescent protein; MEF, mouse embryonic fibroblast; VDAC2, voltage-dependent anion channel 2.

These finding implicate Ala171 as being a key residue involved in VDAC2 interaction with BAK and that this interaction could be modulated to affect BAK apoptotic function.

## The VDAC2 β 10–11 loop is in proximity to the BAK hydrophobic groove

DMS implicated A171 on the VDAC2 β10–11 loop (equivalent to A172 in mouse VDAC2) as an interfacing residue with BAK. We next employed cysteine cross-linking to investigate residues on BAK that interact with mmVDAC2 A172. A172 was mutated to cysteine (VDAC2$^{A172C}$) on a mouse VDAC2$^{\Delta Cys}$ background and stably expressed in $Bak^{-/-}Bax^{-/-}Vdac2^{-/-}$ MEFs. As a control, we expressed VDAC2 with a single cysteine introduced on the opposite side of the β-barrel (VDAC2$^{S58C}$) (S2 Fig). Following stable expression of selected BAK single cysteine variants (Fig 2A and 2B), we treated membrane fractions with the oxidant copper phenanthroline (CuPhe) to induce disulphide bond formation and analysed the linkage adducts by non-reducing SDS-PAGE. Efficient disulphide-linkage was detected between VDAC2$^{A172C}$ and BAK$^{R88C}$/BAK$^{Y89C}$, and to a lesser extent BAK$^{N124C}$ (Fig 2C). No detectable linkage was observed between VDAC2$^{A172C}$ and BAK$^{\Delta Cys}$, BAK$^{G4C}$, BAK$^{R156C}$, or BAK$^{D160C}$ (Fig 2C). None of the BAK variants exhibited specific linkage to VDAC2$^{S58C}$ (Fig 2C). These data suggested that BAK hydrophobic groove residues are proximal to the VDAC2 β10–11 loop.

To further explore residues on BAK that may interface with VDAC2, we then tested residues on the periphery of the groove in which R88, Y89, and N124 are located, including in the α2, 3, 4, and 8 (Fig 2A and 2B). Interestingly, the BAK variants D84C, I85C, E92C, S117C, and S121C were all able to crosslink with VDAC2$^{A172C}$, but I123C, which orientates away from N124, and A179C which is further from the centre of the groove were both unable to do so (Fig 2D). These cross-linking data (Fig 2E) identified a surface on BAK formed by its α3, 4, and 5 that interfaces the cytosol-exposed β10–11 loop on VDAC2 (Fig 3).

## Obstructive labelling of the BAK hydrophobic groove dissociates BAK from VDAC2 and the mitochondrial outer membrane

Cells lacking VDAC2 are primed to undergo BAK-mediated apoptosis, although counter-intuitively, mitochondrial targeting of BAK is impaired [11]. Previous mutagenesis studies have suggested that the BAK BH3 domain mutant L76E disrupts the BAK–VDAC2 interaction while the BAK BH1 domain triple mutant W123A/G124E/R125A reduces it [4]. While these mutagenesis studies, and our DMS and linkage analysis informed residues involved in the BAK–VDAC2 interaction, they did not inform the consequence of acute disruption of the interaction of BAK with mitochondrial VDAC2. To determine this, we employed an obstructive chemical labelling approach that we have previously used to decipher how BAK is activated by BH3-only proteins [17], to investigate the effect of acute disruption of the BAK–VDAC2 interaction. We designed BAK variants that introduce a single cysteine residues at positions on the exposed surface of pre-activated BAK (Fig 2A and 2B) [18] that have been implicated in BAK oligomerisation: the N-terminus (G4), the BH3 domain (R88), the hydrophobic groove (N124), and the α6 helix (R156) [19,20]. To ensure that any observed effect was due to labelling of cysteine introduced into BAK, as opposed to labelling of cysteines in VDAC2, we engineered an HA-tagged VDAC2 mutant that lacked all cysteines (HA-VDAC2$^{\Delta Cys}$). Mitochondria-enriched membrane fractions from $Bak^{-/-}Bax^{-/-}Vdac2^{-/-}$ mouse embryonic fibroblasts (MEFs) expressing HA-VDAC2$^{\Delta Cys}$ and BAK variants were isolated and incubated with 5 kDa PEG-maleimide (PEG-mal) to label exposed cysteines on BAK. Digitonin-solubilised membrane fractions were then subjected to immunoprecipitation of HA-tagged VDAC2 to assess the interaction of BAK. Following treatment of membrane fractions with PEG-maleimide, a Cys-null variant of BAK (BAK$^{\Delta Cys}$) was unaffected in its capacity to coimmunoprecipitate with HA-VDAC2$^{\Delta Cys}$

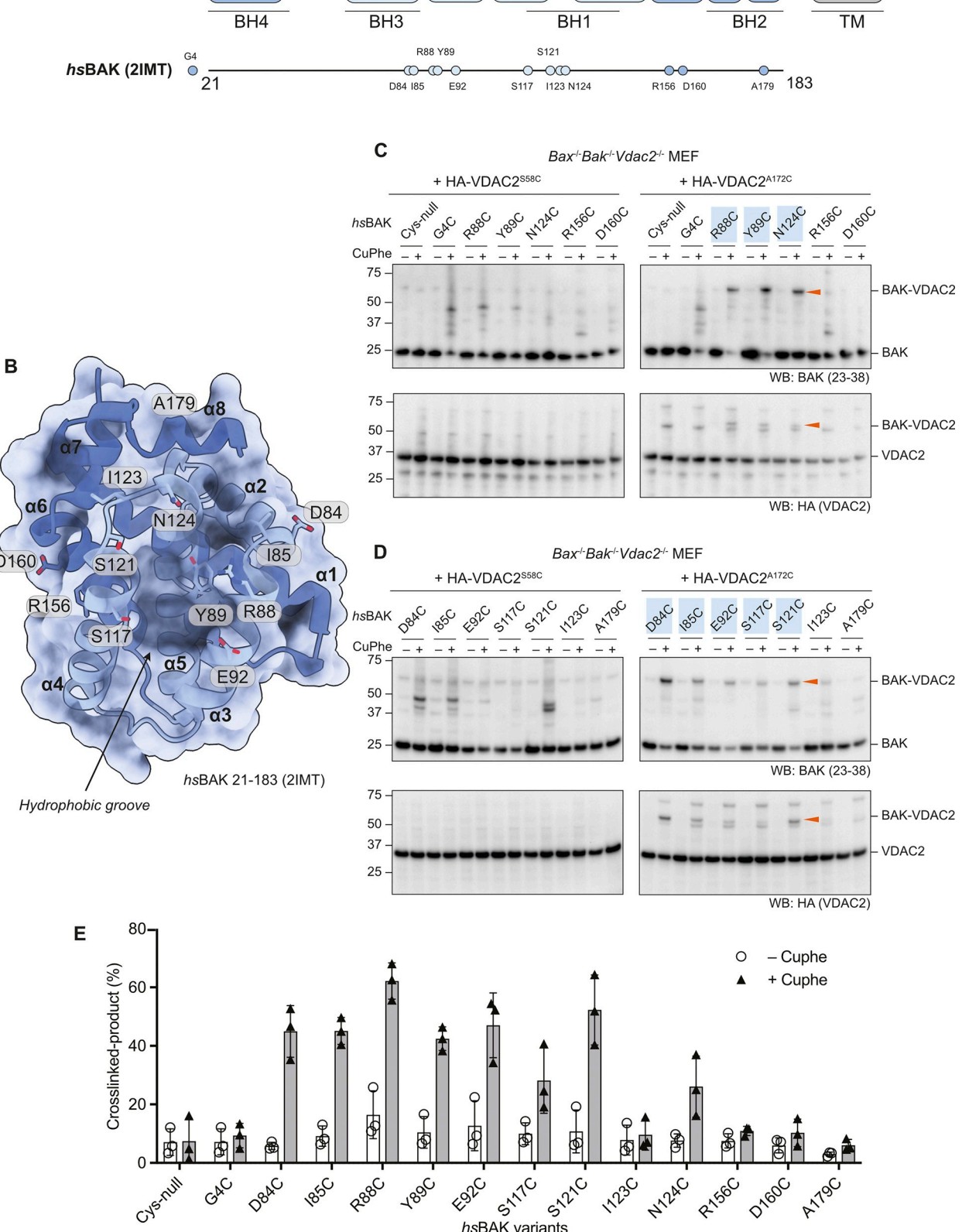

**Fig 2. Cysteine cross-linking identifies BAK residues proximal to VDAC2. (A)** Domain architecture of human BAK. Relative positions of selected single cysteine mutations (on a BAK ΔCys–C14S/C166S background) used in this study are labelled. **(B)** Structure of human BAK (PDB

code: 2IMT) [18] shown in cartoon and surface representations with α helices 2–5 (hydrophobic groove) coloured in light blue. Residues selected for cysteine mutagenesis and cross-linking are shown as sticks. **(C and D)**. Heavy membrane fractions containing mitochondria were isolated from *Bax$^{-/-}$Bak$^{-/-}$Vdac2$^{-/-}$* MEFs ectopically expressing HA-VDAC2$^{S58C}$ or HA-VDAC2$^{A172C}$ (both on a VDAC2 ΔCys background) in combination with different BAK single cysteine mutants. Cysteine cross-linking was induced by incubation with 1 mM CuPhe at 4°C for 30 min and analysed by non-reducing SDS-PAGE. Cross-linked BAK and VDAC2 are denoted by arrowheads. Immunoblots are representative of at least 3 independent experiments. **(E)** Densitometric analysis of CuPhe cross-linking. Cross-linked BAK–VDAC2 (%) was calculated by determining the intensity of cross-linked BAK over the sum intensity of total BAK. Data are mean ± SD of 3 independent experiments. The data underlying this figure is available in S1 Data. MEF, mouse embryonic fibroblast; VDAC2, voltage-dependent anion channel 2.

(Fig 4A), consistent with previous findings that VDAC2$^{ΔCys}$ retains its interaction with BAK [16] and confirming the specificity of this obstructive labelling approach. Labelling of BAK at its N-terminus, distal to the hydrophobic groove (G4C), or at a site on the α6 helix located on the opposite surface of BAK to the hydrophobic groove (R156C) had no effect on the interaction with VDAC2 (Fig 4A). In contrast, labelling at R88C and N124C within the hydrophobic

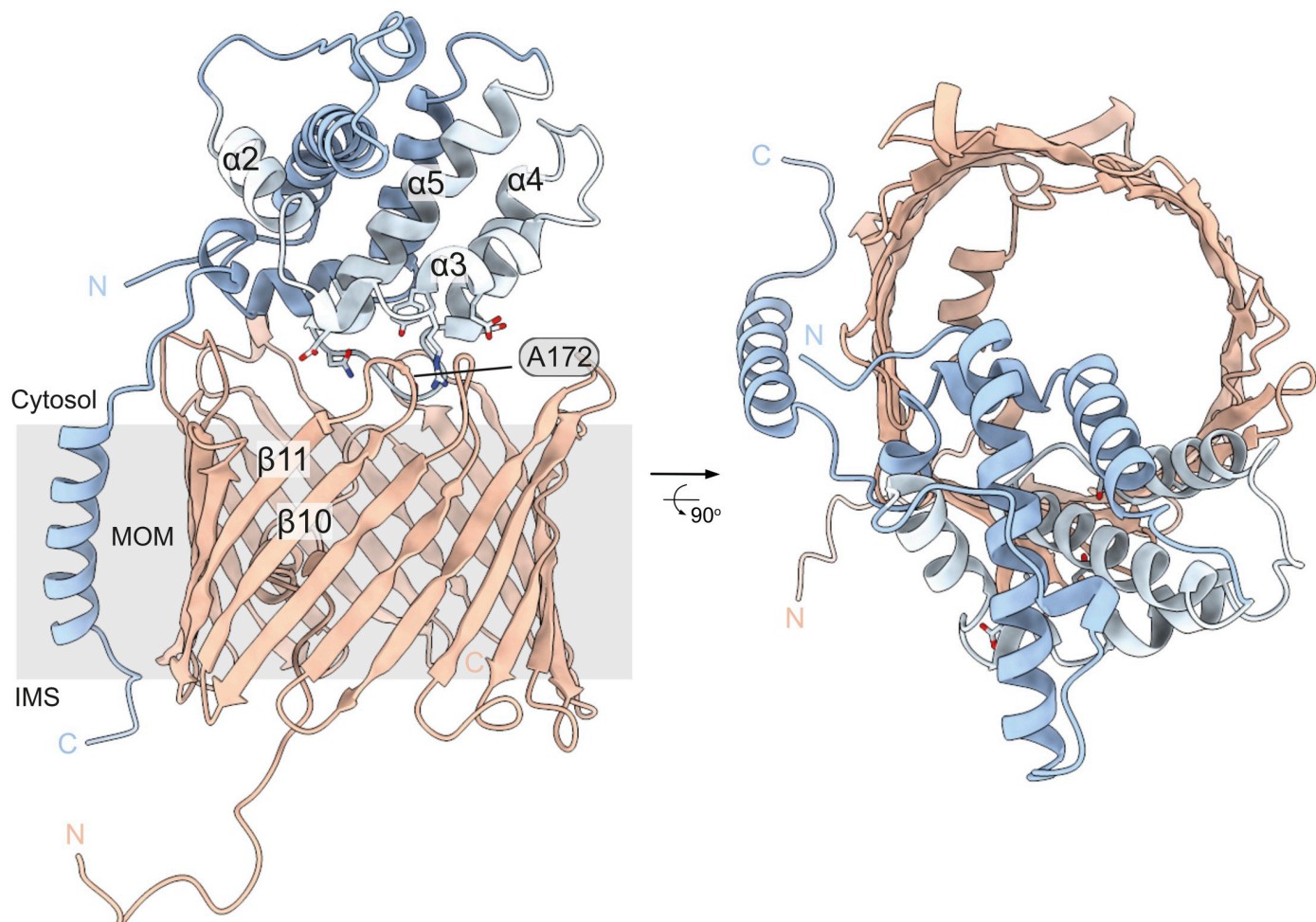

**Fig 3. Hypothetical model of BAK interacting with VDAC2 on the MOM.** Models of the BAK:VDAC2 interaction were generated by multiple rounds of docking initially of the BAK α9 peptide to mVDAC2 β7–β11, followed by docking a minimised BAK α1–8 model to the mVDAC2 solvent exposed A121 loop. Several models were obtained with discrete interaction sites, and each subjected to molecular dynamics simulation. Of the several models generated, this model was selected as the most representative of the cross-linking data. MOM, mitochondrial outer membrane; VDAC2, voltage-dependent anion channel 2.

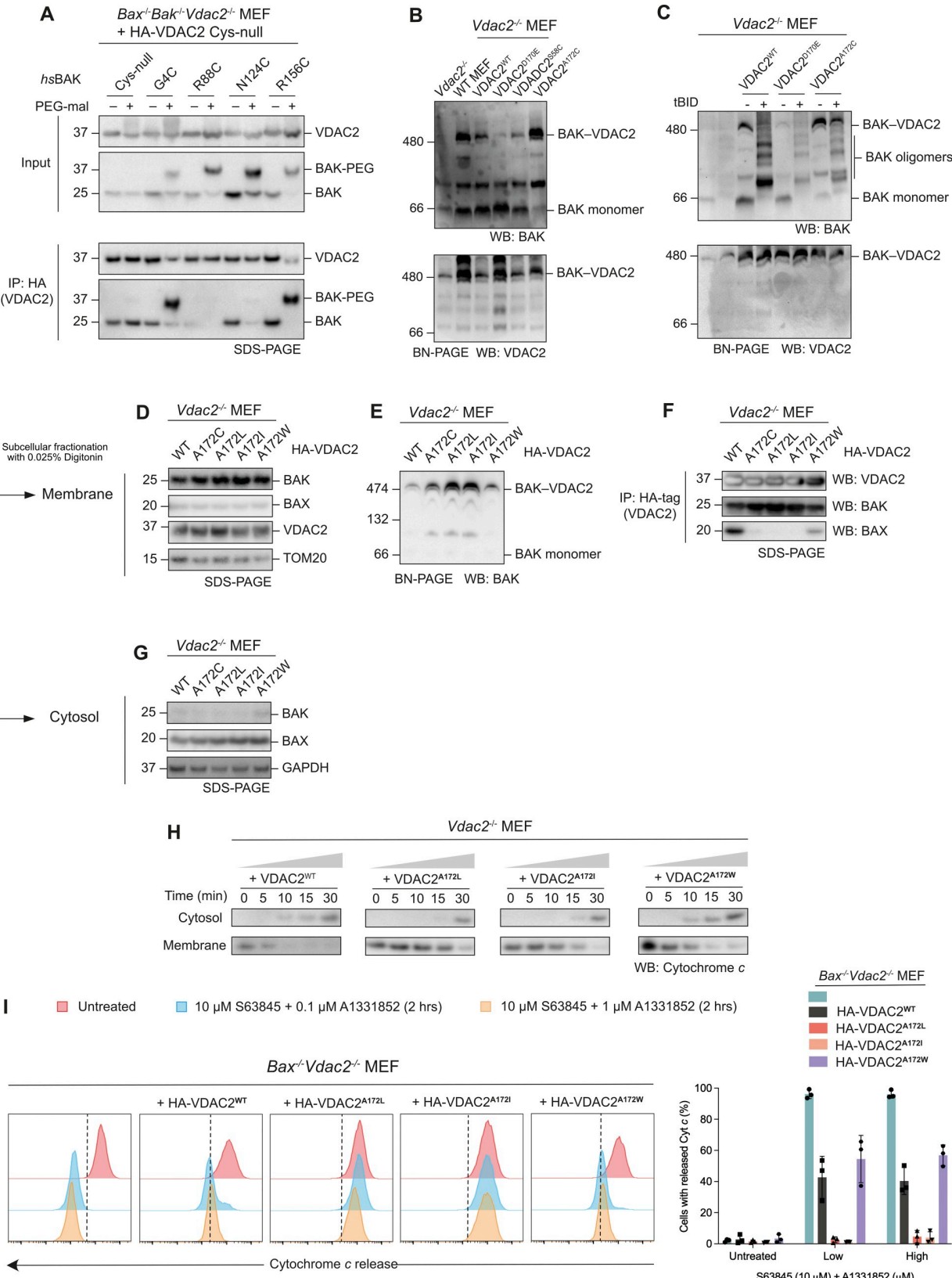

**Fig 4. Mutations of VDAC2 stabilise the BAK interaction. (A)** $Bax^{-/-}Bak^{-/-}Vdac2^{-/-}$ MEF cells expressing HA-VDAC2 ΔCys and BAK ΔCys variants were permeabilised with 0.025% digitonin and treated with PEG-maleimide (5 kDa, 0.5 mM) at room temperature for 30 min. Immunoprecipitations with HA-tag were performed with 1% digitonin solubilised membrane fractions post-labelling. Immunoblots are representative of 3 independent experiments. **(B)** VDAC2 mutants affect the stability of the BAK–VDAC2 complex. Heavy membrane fractions of MEF cells were solubilised by 1% digitonin and analysed by BN-PAGE. Immunoblots are representative of 3 independent experiments. **(C)** BAK activation (dissociation from the VDAC2 complex) induced by tBID (10 nM) was delayed in cells expressing VDAC2$^{A172C}$. Heavy membrane fractions from $Vdac2^{-/-}$ MEFs expressing wild-type VDAC2 or VDAC2 mutants were solubilised with 1% digitonin post-tBID treatment and analysed on BN-PAGE. Immunoblots are representative of 3 independent experiments. **(D–G)** VDAC2$^{A172}$ variants stabilise the interaction with BAK, but destabilise the interaction with BAX. Subcellular fractionation of VDAC2-deficient MEFs ectopically expressing HA-VDAC2$^{A172}$ mutants assessed by subcellular fractionation (D and G), BN-PAGE (E), and immunoprecipitation (F) of solubilised mitochondria. Results are representative of 3 independent experiments. **(H)** VDAC2$^{A172}$ variants that stabilise the interaction with BAK delay the release of cytochrome $c$ as assessed by western blot analysis. Mitochondria-enriched heavy membrane fractions from VDAC2-deficient MEFs expressing HA-VDAC2$^{A172}$ mutants were treated with 10 nM tBID over a time course up to 30 min. Membrane and cytosol fractions were separated and analysed by SDS-PAGE. Immunoblots are representative of 2 independent experiments. **(I)** VDAC2$^{A172}$ variants that stabilise the interaction with BAK reduce the release of Cytochrome $c$ as assessed by flow cytometry. $Bax^{-/-}Vdac2^{-/-}$ MEFs expressing VDAC2$^{A172}$ mutants were treated with BH3 mimetics (10 μm S63845 and 0.1/1 μm A1331852) for 2 h to induce apoptosis. Intracellular flow cytometry was performed using APC-conjugated cytochrome $c$ antibody. Histograms are representative of 3 independent experiments, and % cells with released cytochrome $c$ were plotted, data are mean ± SD. The data including flow cytometry gating strategy underlying this figure is available in S1 Data. FACS data files are available in S2 Data. BN-PAGE, blue native polyacrylamide gel electrophoresis; MEF, mouse embryonic fibroblast; VDAC2, voltage-dependent anion channel 2.

groove of BAK, reduced co-immunoprecipitation with VDAC2 (Fig 4A). Interestingly, mutation of R88 to cysteine disrupted BAK–VDAC2 interaction even in the absence of PEG-mal labelling (Fig 4A). These results suggested that the canonical hydrophobic groove of BAK is involved in its interaction with VDAC2.

## Mutations of VDAC2 stabilise the BAK–VDAC2 complex and limit MOMP

BAK is in complex with VDAC2 on mitochondria in healthy cells [4]. During apoptosis, BAK must dissociate from VDAC2 in order to form dimers and high-order oligomers that permeabilise the MOM [10,20–23]. Therefore, restraining BAK dissociation from VDAC2 might be a way to limit apoptosis. Indeed, it has recently been shown that a small molecule, WEHI-9625, can inhibit BAK-mediated apoptosis by stabilising the BAK–VDAC2 complex [22]. On blue native polyacrylamide gel electrophoresis (BN-PAGE), BAK forms a multi-protein complex with VDAC2, which also includes VDAC1 and VDAC3 [11]. This complex is absent in $Vdac2^{-/-}$ MEF cells and is a destabilised in $Vdac1^{-/-}$ or $Vdac3^{-/-}$ MEFs [11]. However, direct interaction has only been observed between BAK and VDAC2 [4,12]. VDAC1 and 3 do not appear to bind BAK directly and might instead serve as binding partners of VDAC2 within this complex.

To test whether the stability of this BAK–VDAC complex can be affected by single point mutations in VDAC2, we first expressed VDAC2$^{D170E}$, which has been previously reported to disrupt the BAK–VDAC2 interaction [16]. Consistent with published findings, re-expression of VDAC2$^{D170E}$ in $Vdac2^{-/-}$ MEFs did not restore the formation of the BAK–VDAC2 complex (Fig 4B). We found that the expression of VDAC2$^{A172C}$ increased the stability of the BAK–VDAC2 complex compared with wild-type VDAC2 as evidenced by the enriched complex and the virtual absence of monomeric BAK on BN-PAGE (Fig 4B), consistent with our DMS data that indicated that cells expressing this VDAC2 mutant also displayed increased BAK levels overall due to stabilisation at the MOM (Fig 1B and 1C).

BH3-only proteins such as BIM or caspase-8-cleaved BID (truncated BID, tBID) dissociate BAK from its interaction with VDAC2. In this scenario, BAK maintains its localisation at the MOM where it undergoes conformational change and oligomerisation leading to MOMP [10,22]. Whether BH3-only proteins compete with VDAC2 for a shared binding site on BAK, or whether induced changes in BAK conformation allosterically promote BAK–VDAC2

dissociation is unclear. Given its apparent enhanced stability on BN-PAGE, we hypothesised that the stabilised BAK–VDAC2$^{A172C}$ complex would be more resistant to dissociation induced by BH3-only proteins. Membrane fractions isolated from *Vdac2$^{-/-}$* MEFs expressing VDAC2 variants were incubated with recombinant tBID and the BAK–VDAC2 complex assessed on BN-PAGE. Following tBID treatment, BAK completely dissociated from wild-type VDAC2 to form homo-oligomers (Fig 4C). The same was observed for cells expressing VDAC2$^{D170E}$ (Fig 4C). However, tBID was less able to promote dissociation of BAK from VDAC2$^{A172C}$ and consequently BAK activation and oligomerisation were limited (Fig 4C).

We next tested the effect of other VDAC2$^{A172}$ variants, focussing on mutants that either increased (VDAC2$^{A172C/I/L}$) or decreased (VDAC2$^{A172W}$) BAK levels in our DMS assay. We expressed these VDAC2 variants in *Vdac2$^{-/-}$* MEFs and *Bax$^{-/-}$Vdac2$^{-/-}$* MEFs and assessed BAK subcellular localisation by both SDS-PAGE and BN-PAGE (Figs 4D–4G, S3A, and S3B). In all cell lines tested, BAK was able to localise to the mitochondria (Figs 4D, 4G, and S3B), suggesting that these mutants could still interact with BAK to mediate its mitochondrial targeting. BN-PAGE showed that the BAK–VDAC2 complex was stabilised by VDAC2$^{A172C/I/L}$ (Figs 4E and S3B). We additionally performed immunoprecipitation with the HA-tag on VDAC2 and found that more BAK co-precipitated with VDAC2$^{A172C/I/L}$ variants further suggesting the stabilisation of the BAK–VDAC2 complex by these VDAC2 variants (Fig 4F).

To test if this stabilisation impacted BAK's ability to mediate MOMP and cytochrome *c* release, supernatant and membrane fractions were fractionated following treatment of mitochondria-enriched heavy membrane with tBID (Fig 4H) in cell lines expressing wild-type VDAC2 or VDAC2$^{A172I/L/W}$. Cytochrome *c* release was delayed from mitochondria isolated from cells expressing VDAC2$^{A172I}$ or VDAC2$^{A172L}$, suggesting that a single point mutation on VDAC2 is sufficient to stabilise the BAK–VDAC2 complex and restrain BAK's capacity to mediate MOMP (Fig 4H). As BAK is constitutively mitochondrial and BAX is predominantly cytosolic argues that BAK is likely the dominant effector of MOMP in these mitochondrial assays. Moreover, VDAC2$^{A172I/L}$ mutation caused BAX to further dissociate from mitochondria, at least in cells lacking BAK (S3C Fig), consistent with our previous report that BAX requires either VDAC2 or BAK to efficiently target mitochondria [10]. However, we could not exclude that residual mitochondrial BAX contributed to cytochrome *c* release in these assays. Hence, to assess the impact of mutations to VDAC2$^{A172}$ in the complete absence of BAX, we assessed cytochrome *c* release in *Bax$^{-/-}$Vdac2$^{-/-}$* MEFs, reconstituted with VDAC2 variants, treated with BH3 mimetics (Figs 4I, S3A, and S4). When BAK is the sole apoptosis effector, constraining mutations of VDAC2$^{A172I/L}$ mutation inhibited cytochrome *c* release (Fig 4I).

## Stabilisation of the BAK–VDAC2 complex is sufficient to inhibit apoptosis

To understand the consequences of stabilising the BAK–VDAC2 interaction and the influence of this on cell fate, cell death assays were following treatment with BH3-mimetic compounds to specifically induce intrinsic apoptosis in *Bax$^{-/-}$* MEFs to isolate BAK-apoptotic function [24–26]. *Bax$^{-/-}$Vdac2$^{-/-}$* MEFs expressing wild-type VDAC2 or VDAC2$^{A172W}$ could still undergo apoptosis in response to combined treatment with BH3 mimetics S63845 (MCL-1 inhibitor) and A1131852 (BCL-xL inhibitor) (Fig 5A). In contrast, MEFs expressing VDAC2$^{A172L}$ or VDAC2$^{A172I}$ were highly resistant to combined BH3 mimetics. Consistently, in BH3 mimetic-treated cells expressing VDAC2$^{A172L/I}$, BAK remained in an inactive conformation based on binding of the conformation-specific antibody G317-2 [27], but was readily activated in cells expressing wild-type VDAC2 or VDAC2$^{A172W}$ (Figs 5B and S5). This suggested that dissociation of the BAK–VDAC2 complex is a key step in BAK-mediated apoptosis, and that preventing BAK dissociation is sufficient to inhibit BAK activation and subsequent BAK-induced apoptosis.

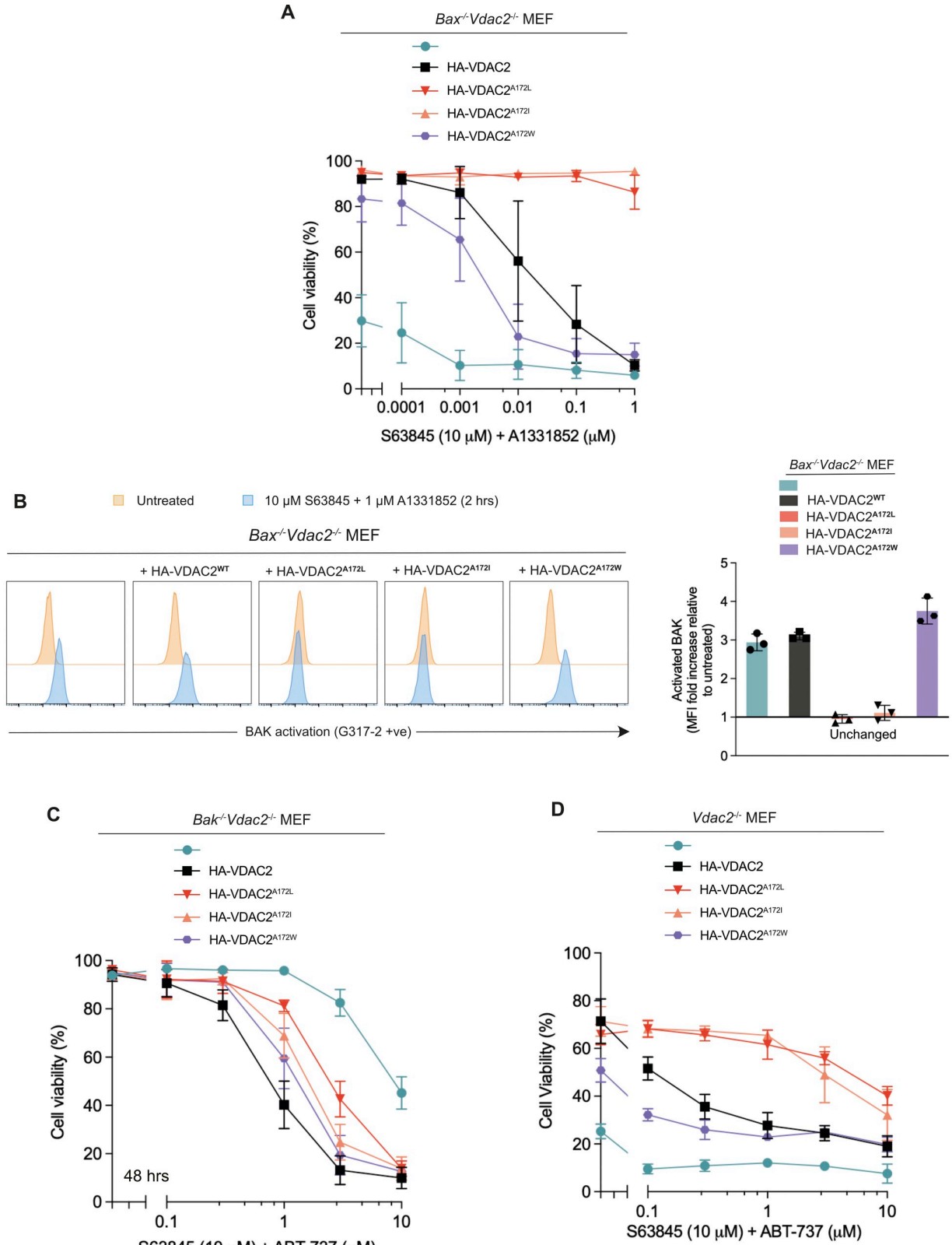

**Fig 5. Stabilisation of the BAK–VDAC2 interaction limits apoptosis. (A)** Stabilisation of the BAK–VDAC2 interaction inhibits apoptosis. $Bax^{-/-}Vdac2^{-/-}$ MEFs ectopically expressing VDAC2$^{A172}$ variants were treated with 10 μm S63845 (MCL-1 inhibitor) and escalating doses of

A1331852 (BCL-xL inhibitor) for 24 h. Cells were stained with PI and analysed by flow cytometry. Data are mean ± SD of 3 independent experiments. **(B)** Stabilisation of the BAK–VDAC2 interaction inhibits BAK activation. *Bax⁻/⁻ Vdac2⁻/⁻* MEFs ectopically expressing VDAC2$^{A172}$ variants were treated with BH3 mimetics (10 µm S63845 and 1 µm A1331852) for 2 h. BAK activation was assessed by intracellular flow cytometry with the BAK conformation-specific antibody G317-2. Representative histograms are shown in the left panel, fold increase in MFI plotted in the right panel. Data in the right panel are mean ± SD of 3 independent experiments. **(C)** In the absence of BAK, destabilisation of the BAK–VDAC2 interaction inhibits apoptosis. *Bak⁻/⁻ Vdac2⁻/⁻* MEFs ectopically expressing VDAC2$^{A172}$ variants were treated with 10 µm S63845 (MCL-1 inhibitor) and escalating doses of A1331852 (BCL-xL inhibitor) for 24 h. Cells were stained with PI and analysed by flow cytometry. **(D)** *Vdac2⁻/⁻* MEFs expressing VDAC2A172 variants were treated and assessed as in (C). Data are mean ± SD of 3 independent experiments. The data including flow cytometry gating strategy underlying this figure is available in S1 Data. FACS data files are available in S2 Data. MEF, mouse embryonic fibroblast; MFI, mean fluorescence intensity; PI, propidium iodide; VDAC2, voltage-dependent anion channel 2.

## VDAC2$^{A172}$ also influences BAX–VDAC2 interaction and BAX-mediated apoptosis

VDAC2 has been implicated to have contrasting effects on BAK and BAX activity, as it can inhibit BAK-mediated apoptosis, but is required for BAX-mediated apoptosis [4,11]. This has been attributed to the differing subcellular localisation of BAX and BAK [11,28]. Unlike BAK, BAX is mainly cytosolic with redistribution from the cytosol to the MOM required for its apoptotic activity [29,30]. BAX utilises VDAC2 to target mitochondria because BAX cannot localise to mitochondria in cells devoid of VDAC2, and therefore its ability to kill cells is impaired [10,11]. As mutation of VDAC2$^{A172}$ influenced interaction with BAK, we wanted to understand if this residue was also involved in VDAC2's interaction with BAX. We assessed the formation of the BAX–VDAC2 mitochondrial complex in *Vdac2⁻/⁻* MEFs expressing VDAC2$^{A172}$ variants. Surprisingly, mutations to VDAC2 that stabilised its interaction with BAK (VDAC2$^{A172L/I}$), actually destabilised its interaction with BAX as assessed by co-immunoprecipitation (Fig 4F) or BN-PAGE (S3C Fig). That the VDAC2-BAX complex was destabilised by VDAC2$^{A172L/I}$ mutation in *Bak⁻/⁻ Vdac2⁻/⁻* MEFs ruled out an effect of competitive binding of BAK and BAX (S3C Fig). In *Bak⁻/⁻ Vdac2⁻/⁻* MEFs expressing VDAC2$^{A172L/I}$, BAX was less able to mediate cell death than in cells expressing wild-type VDAC2 or VDAC2$^{A172W}$ (Fig 5C). Hence, while specific mutation of VDAC2$^{A172}$ had opposing effects on VDAC2 interaction with BAX and BAK, the consequence for BAX or BAK-mediated killing was similar, likely due to their respective cytosolic and mitochondrial localisation.

Most cells express both BAX and BAK. To assess if stabilising the BAK–VDAC2 interaction could inhibit the death of cells that express both BAK and BAX, we investigated cell death in *Vdac2⁻/⁻* MEFs. Cells reconstituted with VDAC2$^{A172I/L}$ variants were significantly more resistant to apoptosis induced by BH3 mimetics compared with *Vdac2⁻/⁻* cells expressing wild-type VDAC2 (Fig 5D). In contrast, cells expressing VDAC2$^{A172W}$ were more sensitive to BH3 mimetics (Fig 5D). This suggested that under certain circumstances, limiting BAK apoptotic function by stabilising its interaction with VDAC2 may be sufficient to inhibit cell death even in the presence of functional BAX.

## VDAC2$^{A172W}$ mutation prevents BAK-inhibition by WEHI-9625

WEHI-9625 is a small molecule that binds VDAC2 to inhibit mouse BAK-mediated apoptosis by stabilising the BAK–VDAC2 interaction [22]. However, the binding site for WEHI-9625 on VDAC2 is poorly defined. As VDAC2$^{A172L/I}$ also stabilised the BAK–VDAC2 interaction and rendered cells resistant to death, thereby mimicking WEHI-9625 activity, we hypothesised that VDAC2$^{A172}$ may be involved in WEHI-9625 binding. To test this, we treated *Bax⁻/⁻ Vdac2⁻/⁻* MEFs expressing wild-type VDAC2 or VDAC2$^{A172W}$, both of which respond to BH3 mimetics (Fig 6A), with an approximate EC$_{50}$ concentration of combined BH3 mimetics together with an increasing concentration of WEHI-9625. While WEHI-9625 was able to inhibit the death of

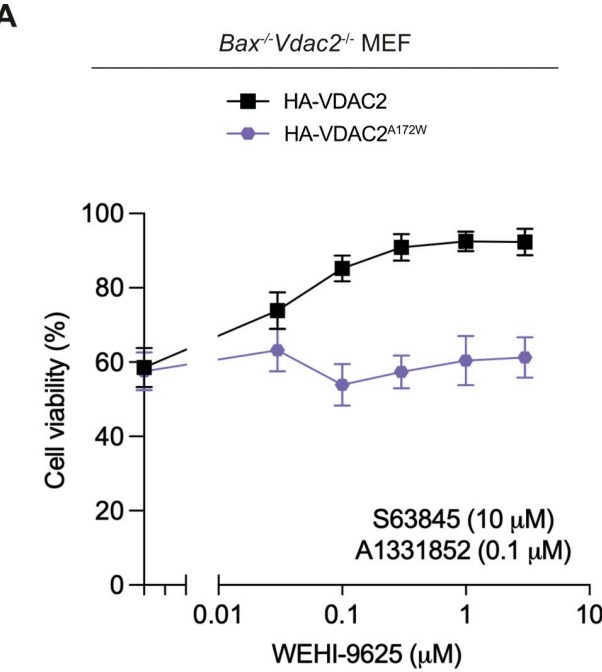

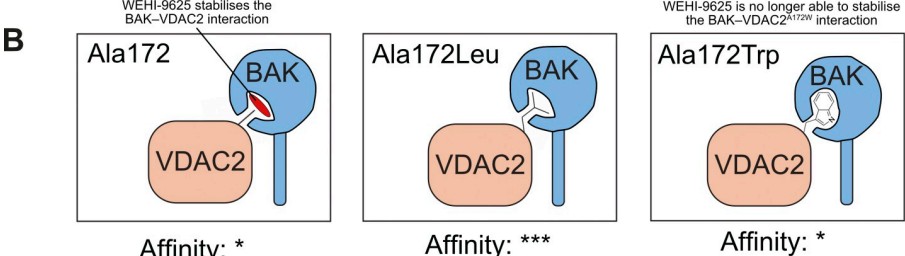

**Fig 6. VDAC2$^{A172W}$ mutation limits WEHI-9625 function. (A)** BAX/VDAC2-deficient MEFs expressing HA-VDAC2 or HA-VDAC2$^{A172W}$ were treated with an approximate EC50 dose of BH3-mimetics (10 μm S63845 and 0.1 μm A1331852) and increasing doses of WEHI-9625 as indicated. Cell viability was determined by measuring PI negative cells using flow cytometry. Data are mean ± SD of 3 independent experiments. **(B)** Hypothetical model for how VDAC2$^{A172}$ variants may affect BAK stability on the MOM. WEHI-9625 and VDAC2$^{A172I/L}$ stabilise the BAK–VDAC2 interaction such that BAK dissociation from VDAC2, and ensuing apoptotic activity, is restricted. In cells expressing VDAC2$^{A172W}$, WEHI-9625 is no longer able to stabilise the BAK–VDAC2. A possible molecular explanation for these observations is that VDAC2$^{A172}$ only partially occupies the region of the hydrophobic groove of BAK identified by our cross linking. WEHI-9625, and medium-sized hydrophobic side chains, may exploit the imperfect fit of A172, enhancing interaction by providing greater surface complementarity. The larger bulk of tryptophan at this position may destabilise the interaction relative to isoleucine or leucine through less ideal surface complementarity, at the same time restricting access to this region for WEHI-9625. The data including flow cytometry gating strategy underlying this figure is available in S1 Data. FACS data files are available in S2 Data. MEF, mouse embryonic fibroblast; MOM, mitochondrial outer membrane; PI, propidium iodide; VDAC2, voltage-dependent anion channel 2.

cells expressing wild-type VDAC2, it was unable to inhibit BAK apoptotic function in cells expressing VDAC2$^{A172W}$ (Fig 6A). This suggests that VDAC2$^{A172}$ is involved in the binding site of WEHI-9625 within the BAK–VDAC2 complex, and that mutation at this site to tryptophan restricts compound binding (Fig 6B).

## Discussion

VDAC2 facilitates the mitochondrial localisation of BAK where it acts to mediate apoptosis [10,16]. However, when BAK is on the MOM, VDAC2 sequesters BAK [4]. To drive apoptosis, BAK must dissociate from VDAC2 to oligomerise and form the apoptotic pore [4]. Here, we have sought to map the interface between VDAC2 and BAK and explore the consequence of modulating this interaction. We show that the VDAC2-complex can be acutely disrupted by obstructive labelling of the BAK hydrophobic groove, that mutation to A172 within the VDAC2 β10–11 loop affects BAK localisation and the stability of the VDAC2–BAK complex, and that residues in the groove of BAK can be cross-linked to A172. This provides strong evidence that VDAC2 A172 forms part of an interaction interface within the VDAC2–BAK complex. Previously, we showed that a modified BH3 peptide that could bind the BAK groove with high affinity and displace BAK from the VDAC2 complex [21]. However, we did not know if this was due to a direct competition for the BAK groove or a result of allosteric effects due to the groove opening upon BH3 binding. Our results here show that the BAK groove does indeed bind directly to VDAC2 and that BH3 proteins likely compete with VDAC2 for the groove, explaining our previous observations.

The hydrophobic groove of BCL-2 family proteins is the primary site of interaction with other BCL-2 family member proteins. However, in those cases the interaction interface involves an amphipathic helix, the BH3 domain of the partner, binding into the groove on the BCL-2 fold [31–33]. This interaction is mediated by at least 4 hydrophobic residues on consecutive turns of the helix interacting with hydrophobic pockets within the groove, and a conserved salt bridge between an aspartate on the helix and an arginine in the groove [33]. The interaction topology adopted by the VDAC2 β10–11 loop with the BAK hydrophobic groove interface is unlikely to be similar, primarily because the VDAC2 β10–11 loop is only 6 amino acids so not long enough to form a helix of 4 consecutive turns (each turn in an α helix requires 4 residues). It is more likely that this region of VDAC2 instead inserts into the hydrophobic groove as a loop. Recently, a structure of p53 bound to the pro-survival family member BCL-2 has been reported as having such an interaction [34]. However, to our knowledge, such an interface has not been described for a pro-apoptotic BCL-2 executioner family member.

Interestingly, we observe that the type of substitution made to VDAC2[A172] can have variable impact on the stability of the VDAC2–BAK complex. Mutations to medium-sized hydrophobic sidechains (Ile, Leu, Cys) stabilise the complex, while mutation to a large hydrophobic residue (Trp) is less able to restrain BAK. It is notable that generally the hydrophobic residues on BH3 domains that interact with BCL-2 family hydrophobic grooves are medium-sized, particularly the h2 position which is invariably a leucine. It is possible that the medium-sized mutations to VDAC2 A172 mimic such an interaction to increase the affinity for the groove, thus limiting competition by activating BH3 domains. Alternatively, mutation to the larger tryptophan may not be as readily accommodated, thus reducing its affinity. Mutation from Leu to Ala at the BIM BH3 and BID BH3 h2 positions significantly impairs binding to BAX [35] and BAK [31], respectively. However, in our view this does not necessarily rule out the h2 interacting pocket as being the site of interaction for VDAC2 A172. Such a substitution is likely to be less tolerated in the context of a BH3 helix where backbone conformers are restricted, to maintain hydrogen-bonding in the helix, compared with an inserted flexible loop that can adopt various orientations.

The observation that mutation of VDAC A172 to tryptophan renders cells less responsive to the BAK inhibitor WEHI-9625, suggests that this interface is also the likely site of interaction for this compound. It is not known precisely how WEHI-9625 functions to limit mouse BAK apoptotic activity other than it appears to stabilise the BAK–VDAC2 complex. Our

results suggests that this molecule takes advantage of a pocket within the BAK–VDAC2 complex. We hypothesise that when this pocket is occupied by a medium-sized sidechain such as leucine, isoleucine, or cysteine, it phenocopies the stabilising effect of WEHI-9625. However, when A172 is mutated to a large sidechain such as tryptophan, it can compete with WEHI-9625 for BAK pocket binding, and because the tryptophan does not stabilise the complex an increase in killing is observed. Structures of the BAK-VDAC2 complex with compound bound are ultimately required to resolve this mechanism.

That mutation to VDAC2$^{A172}$ impacts both BAX and BAK activity suggests that this site of interaction is conserved for these 2 proteins. When VDAC2$^{A172}$ is mutated to tryptophan, it decreases the stability of the interaction with BAK, and more BAX becomes associated with VDAC2. Conversely, mutation of VDAC2$^{A172}$ to Leu/Ile had opposing effects on VDAC2 interaction with these 2 proteins—destabilising interactions with BAX, but stabilising interactions with BAK. For BAX this results in reduced recruitment to the mitochondria, especially when BAK is absent, for BAK it results in less freed protein on the mitochondria. In both scenarios, there is a reduction of free BAX and/or BAK on the mitochondria available to induce MOMP, and therefore, the ultimate consequence on cell death is the same. Excitingly, this suggests that this site on VDAC2 can be potentially targeted to inhibit apoptosis in cells expressing both BAK and BAX. While this site on VDAC2 mediates interactions with both BAK and BAX, our data suggest that the effectiveness of targeting this site on VDAC2 to inhibit cell death will likely be more profound in cells, or with apoptotic stimuli, that predominantly rely on BAK.

While it is well established that VDAC2 interacts with BAK and regulates BAK apoptotic activity [4,9,10,12,16], the precise nature of this interaction has remained unclear. Here, we have mapped a new interaction interface between VDAC2 and BAK, providing insight into the nature of this complex at the MOM. Excessive apoptosis is implicated in contributing to a range of disease states including autoimmune disease and acute or chronic neurodegenerative conditions, stabilising the VDAC2–BAK interaction may be a potential therapeutic strategy for inhibiting apoptosis in such settings.

## Materials and methods

### Cell culture and retroviral infection

MEFs were immortalised by SV-40 transformation and maintained in Dulbecco's Modified Eagle Medium (DMEM) supplemented with 8% fetal bovine serum (FBS), 250 μm asparagine, and 50 μm 2-Mercaptoethanol (2-ME) in 10% $CO_2$ and at 37°C. Cells used were routinely tested to confirm mycoplasma negativity with MycoAlert kit (Lonza, Basal, Switzerland).

BAK and VDAC2 constructs were stably expressed in MEFs by retroviral infection. Retroviral expression vectors pMX-IG, pMXs-IH, and pMXs-IP were first introduced into Phoenix cells by FuGENE6 (Promega, Wisconsin, United States of America) transfection. Virus-containing supernatants were filtered, supplemented with 4 μg/ml polybrene (Sigma-Aldrich, Missouri, USA) and used to infect MEFs by spin infection (2,500 rcf centrifugation at 32°C for 45 min). Cells stably expressing constructs were selected by culturing with 300 μg/ml hygromycin (Thermo Fisher) or 2 μg/ml puromycin (Sigma-Aldrich), or by FACS-sorting of GFP-positive.

### Subcellular fractionation and cytochrome *c* release assay

MEF cells were harvested and washed with PBS, followed by permeabilisation with 0.025% digitonin (w/v) in modified egg lysis buffer (MELB, 20 mM HEPES (pH 7.4), 100 mM sucrose, 2.5 mM $MgCl_2$, 100 mM KCl) supplemented with cOmplete protease inhibitors (Roche) for 5 min on ice. Heavy membrane fractions were collected by centrifugation (18,000 rcf for 5 min

at 4˚C) and resuspended in MELB supplemented with 10 nM tBID for 30 min at 30˚C. Centrifugation (18,000 rcf for 5 min at 4˚C) were then performed to separate membrane (insoluble pellet) and cytosol (supernatant) fractions. For SDS-PAGE running, cytosol fractions were supplemented with 2× reducing sample buffer while membrane fractions were resuspended in 2 volumes of 1× reducing sample buffer.

### PEG-maleimide labelling of BAK

Membrane fractions were resuspended and incubated in MELB supplemented with 0.5 mM PEG-maleimide (5 kDa, Sigma-Aldrich) and cOmplete protease inhibitors (Roche) at room temperature for 30 min. The reaction was then quenched by addition of *n*-ethylmaleimide at a final concentration of 20 mM. Proteins were solubilised in 1% w/v digitonin for 30 min at 4˚C prior to immunoprecipitation of HA-VDAC2 with anti-HA beads (Thermo Fisher) for 1 h at 4˚C. Proteins were eluted from the beads by boiling for 5 min in SDS-PAGE sample buffer prior to analysis on SDS-PAGE.

### Disulphide linkage of cysteine mutants on mitochondria

Cells were harvested, washed with PBS, and permeabilised at $1 \times 10^7$ cells/ml in MELB with cOmplete protease inhibitors (Roche) and 0.025% digitonin (w/v) for 10 min on ice. Membranes were collected by centrifugation (18,000 rcf for 5 min at 4˚C) and resuspended in MELB with 1 mM copper(II)(1,10-phenanthroline)$^3$ (CuPhe) for 30 min at 4˚C. The reactions were quenched by addition of EDTA to a final concentration of 20 mM and centrifuged at 18,000 rcf for 5 min at 4˚C to remove the supernatant. Pellets were resuspended in 2 volumes of 1× non-reducing SDS-sample buffer and assessed by SDS-PAGE.

### Cell viability assay by flow cytometry

Cells were seeded in 24-well plates overnight and then treated with BH3 mimetics as indicated. After treatment for 24 h or 48 h, all cells were collected and resuspended in KDA-BSS buffer supplemented with 2.5 μg/ml propidium iodide (PI) (Sigma-Aldrich). Samples were analysed by FACS on an LSR-II flow cytometer (BD Biosciences).

### BAK activation and cytochrome *c* release assay by intracellular flow cytometry

Cells were harvested followed by pretreatment with 20 μM QVD for 30 min and BH3 mimetics (10 μM S63845 and 1 μM A1331852) for 2 h. Cells were fixed and permeabilised using the eBioscience cell fixation and permeabilisation kit (Thermo Fisher) according to manufacturer's instructions. Activated BAK was detected by the BAK conformation-specific antibody G317-2 (1:100 dilution in permeabilisation buffer, BD Pharmingen) and subsequently phycoerythrin (PE)-conjugated anti-mouse antibody (1:200 dilution in permeabilisation buffer SouthernBiotech). Samples were analysed by FACS on an LSR-II flow cytometer (BD Biosciences).

To assess cytochrome *c* release, cells were permeabilised in MELB buffer (20 mM HEPES (pH 7.5), 100 mM sucrose, 2.5 mM MgCl$_2$, 100 mM KCl) supplemented with 0.025% digitonin on ice for 30 min. Cytosol fractions were removed and cells were fixed in fixation buffer. Fixed cells were incubated with APC-conjugated cytochrome *c* antibody (1: 50 dilution in permeabilisation buffer, Miltenyi Biotec) and analysed by FACS on an LSR-II flow cytometer (BD Biosciences).

## SDS-PAGE, Blue Native-PAGE, and immunoblotting

For SDS-PAGE running, 2× SDS sample buffer was added to the supernatant and an equal volume of 1× SDS sample buffer was used to resuspend the membrane fractions. For Blue Native-PAGE, membranes were resuspended in 20 mM Bis-Tris (pH 7.4), 50 mM NaCl, 10% glycerol (v/v) supplemented with cOmplete protease inhibitors (Roche), 5 μm DTT, and 10% digitonin (w/v) and incubated on ice for 30 min. Soluble membrane proteins were obtained by centrifugation (18,000 rcf for 5 min at 4°C), supplemented with 10× BN loading dye (5% Coomassie Blue R-250 (Bio-Rad Laboratories, CA) in 500 mM 6-aminohexanoic acid, 100 mM Bis-Tris (pH 7.0)) and run-on BN-PAGE. Gels were transferred to polyvinylidene fluoride (PVDF) membranes (Invitrogen) either by semi-dry transfer using a Trans-Blot Semi-Dry Transfer system (Bio-Rad) or by wet-transfer in Tris-glycine transfer buffer (25 mM Tris (pH 7.4), 192 mM glycine, 20% (v/v) methanol, and 0.037% SDS) at a constant voltage of 30 V for 150 min. Membranes were blocked with 5% (w/v) non-fat milk in Tris-buffered saline-Tween (TBS-T, 20 mM Tris-HCl (pH 7.4), 137 mM NaCl, 0.1% (v/v) Tween-20) at room temperature for 1 h prior to incubation with primary antibody as indicated. Followed washing with TBS-T, membranes were incubated with appropriate horseradish peroxidase-conjugated secondary antibodies and were developed on a ChemiDoc MP (Bio-Rad) machine with enhanced chemiluminescence (ECL) solution (Millipore).

## Deep mutational scanning screens

A plasmid library encoding FLAG-tagged human VDAC2 was constructed in the MSCV-puro retrovirus backbone by randomising individual codons between positions Q165 and F180. A 15-nucleotide barcode was incorporated in the 3′ UTR and associated to the upstream coding substitutions by sequence validation. The final library comprised 4,920 sequence-verified clones linked to unique barcodes, of which 289 corresponded to wild-type or synonymous variants, 247 to premature stop codons, and a median of 11 unique barcodes corresponded to other individual coding substitutions.

The DMS screen to assess the expression level of VDAC2 variants in this library was performed by low MOI (<0.3) retroviral infection into $Mcl1^{-/-}Bax^{-/-}Vdac2^{-/-}$ MEFs at >300X library coverage ($1.5 \times 10^6$ infected cells). Infected cells were selected with 3 μg/ml puromycin, expanded, fixed with methanol, permeabilized, and stained with PE-conjugated anti-FLAG antibody (Clone L5; BioLegend). FLAG$^{high}$ and FLAG$^{low}$ cells were sorted and DNA was extracted with DNeasy columns (Qiagen) for sequencing library preparation. The library was prepared for Illumina sequencing using a two-step PCR amplification protocol as previously described [36]. The primers used for the first PCR step (overhangs in uppercase) were GTGACCTATGAACTCAGGAGTCctggaggccacaaggttg and CTGAGACTTGCACATCG-CAGCggtggatgtggaatgtgtg and 3 μg of genomic DNA from each sample was used as template across replicate PCRs.

The screen to assess the capacity of VDAC2 variants to recruit and stabilise BAK at the MOM was performed similarly, with the following modifications. $Mcl1^{-/-}Bax^{-/-}Bak^{-/-}Vdac2^{-/-}$ MEFs were first engineered to express GFP-BAK by infection with MSCV-EGFP(mm) BAK-IRES-hygro retrovirus, followed by selection with 500 μg/ml hygromycin B. The DMS library was then introduced into these cells as above. Library-transduced cells were expanded, sorted into GFP$^{high}$ and GFP$^{low}$ fractions and used for Illumina sequencing library preparation as above.

## VDAC2-BAK (α1–9) modelling

Details of VDAC2–BAK interaction modelling and molecular dynamics simulations are provided as S1 Methods.

## Supporting information

**S1 Fig. Supplementary data for deep mutational scanning assays. (A)** MCL1/BAX/VDAC2-deficient MEFs engineered by retroviral infection to express wild-type FLAG-VDAC2 or a library of uniquely barcoded FLAG-tagged VDAC2 substitution variants within the β10–11 loop were fixed and stained with PE-conjugated anti-FLAG antibody. The distribution of expression observed in cells expressing VDAC2 variants from the library was broader than for wild-type VDAC2, reflecting variation in expression level for clones within the library. **(B)** BAX/VDAC2-deficient MEFs expressing barcoded FLAG-VDAC2 substitution variants were fixed and stained with PE-conjugated anti-FLAG antibody. FLAGhigh and FLAGlow populations were sorted and used to generate Illumina sequencing libraries in order to identify substitutions that impair or enhance VDAC2 expression. **(C)** Heatmap representation of DMS screen to identify residues within hsVDAC2 β10–11 that influence VDAC2 expression level. Illumina sequencing was performed on the sorted FLAGhigh and FLAGlow populations to quantitate PCR-amplified barcode levels. Mann–Whitney $p$-values were calculated comparing the log2-fold differences for barcodes associated with each coding substitution relative to the barcodes associated with wild-type VDAC2 coding sequence. Data are represented as signed log transformed $p$-values to reflect the direction of change: negative/blue for variants skewed towards the FLAGlow fraction; positive/red for variants skewed towards the FLAGhigh fraction. **(D)** MCL1/BAX/BAK/VDAC2-deficient MEFs were engineered by retroviral infection to express GFP-tagged mouse BAK. In the absence of VDAC2, the expression levels of GFP-BAK are low (green histogram). When FLAG-tagged wild-type human VDAC2 is expressed in these cells, the GFP-fluorescent signal is markedly enhanced (blue histogram). **(E)** MCL1/BAX/BAK/VDAC2-deficient MEFs expressing uniquely barcoded FLAG-tagged VDAC2 substitution variants within the β10–11 loop were sorted into GFPhigh and GFPlow fractions. These were used to generate Illumina sequencing libraries to identify substitutions that impair or enhance the ability of VDAC2 to recruit or stabilise BAK at the MOM (see Fig 1B). The data underlying this figure is available in S1 Data.
(TIF)

**S2 Fig. AlphaFold2 predicted model of mouse VDAC2 (Uniprot Q60930), demonstrating predicted relative positions of A172 and S58 (mouse numbering).** Relates to Fig 2.
(TIF)

**S3 Fig. VDAC2A172L/I mutants stabilise the BAK complex but destabilise the BAX complex (relates to Figs 4 and 5).**
(TIF)

**S4 Fig. Repeats of cytochrome *c* release assay (relates to Fig 4I).**
(TIF)

**S5 Fig. Repeats of BAK activation assay (relates to Fig 5B).**
(TIF)

**S1 Data. The underlying replicate data for the generation of Figs 1B, 1C, 2E, 4I, 5A–5D, 6A, S1C, and flow cytometry gating strategy for Figs 4I, 5A–5D, 6A.**
(XLSX)

**S2 Data. The flow cytometry files for the generation of Figs 4I, 5A–5D, and 6A.**
(ZIP)

**S1 Raw Images. The original uncropped gel images for data in figures.**
(PDF)

**S1 Methods. Appendix supplementary methods.**
(DOCX)

## Author Contributions

**Conceptualization:** Mark F. van Delft, Richard W. Birkinshaw, Peter E. Czabotar, Grant Dewson.

**Data curation:** Zheng Yuan, Mark F. van Delft, Mark Xiang Li, Brian J. Smith, Ruitao Jin, Sitong He, Nicholas A. Smith.

**Formal analysis:** Zheng Yuan, Mark F. van Delft, Mark Xiang Li, Brian J. Smith, Ruitao Jin, Sitong He, Nicholas A. Smith.

**Investigation:** Zheng Yuan, Mark F. van Delft, Mark Xiang Li, Brian J. Smith, Ruitao Jin, Sitong He, Nicholas A. Smith, Richard W. Birkinshaw, Peter E. Czabotar, Grant Dewson.

**Methodology:** Zheng Yuan, Mark F. van Delft, Mark Xiang Li, Brian J. Smith, Ruitao Jin, Sitong He, Nicholas A. Smith, Richard W. Birkinshaw, Peter E. Czabotar, Grant Dewson.

**Resources:** Fransisca Sumardy, David C. S. Huang, Guillaume Lessene, Yelena Khakam.

**Visualization:** Zheng Yuan, Mark F. van Delft, Richard W. Birkinshaw, Peter E. Czabotar, Grant Dewson.

**Writing – original draft:** Zheng Yuan, Mark F. van Delft, Richard W. Birkinshaw, Peter E. Czabotar, Grant Dewson.

**Writing – review & editing:** Zheng Yuan, Mark F. van Delft, Mark Xiang Li, Fransisca Sumardy, Brian J. Smith, David C. S. Huang, Guillaume Lessene, Yelena Khakam, Ruitao Jin, Sitong He, Nicholas A. Smith, Richard W. Birkinshaw, Peter E. Czabotar, Grant Dewson.

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
