## [Editor Report · Decision Letter 0]

11 Jan 2024

Dear Dr Dewson, 

Thank you for submitting your manuscript entitled "Molecular definition of the BAK:VDAC2 interaction as a target to manipulate apoptosis" for consideration as a Research Article by PLOS Biology.

Your manuscript has now been evaluated by the PLOS Biology editorial staff as well as by an academic editor with relevant expertise and I am writing to let you know that we would like to send your submission out for external peer review.

Once your full submission is complete, your paper will undergo a series of checks in preparation for peer review. After your manuscript has passed the checks it will be sent out for review. To provide the metadata for your submission, please Login to Editorial Manager (https://www.editorialmanager.com/pbiology) within two working days, i.e. by Jan 15 2024 11:59PM.

Kind regards,

Ines

--

Ines Alvarez-Garcia, PhD

Senior Editor

PLOS Biology

---

## [Decision Letter · Decision Letter 1]

12 Mar 2024

Dear Dr Dewson,

Thank you for your patience while your manuscript entitled "Molecular definition of the BAK:VDAC2 interaction as a target to manipulate apoptosis" went through peer-review at PLOS Biology. Please also accept my apologies again for the time it has taken us to provide you with a decision. Your manuscript has now been evaluated by the PLOS Biology editors, an Academic Editor with relevant expertise, and by four independent reviewers.

The reviews are attached below. As you will see, the reviewers find the conclusions interesting and significant for the field, but they also raise a few issues that they would like you to address. Reviewer 1 thinks you should show more clearly where the residues at the Bak hydrophobic groove are located, include a structural comparison analysis of Bak in a VDAC2-bound and in BH3-bound forms, and also confirm with other methods that the stabilisation of the BAK-VDAC2 complex interferes with apoptosis. The other reviewers only ask for several clarifications and to consider alternative interpretations of some of the data.

After discussing the reviews with the Academic Editor, we are pleased to offer you the opportunity to address the comments from the reviewers in a revision that we anticipate should not take you very long. We have decided to make optional the requests made by Reviewer 1 regarding the localisation of the residues at the Bak hydrophobic groove and the structural comparison analyses. Although it would be nice to have them, they won't change the main conclusions.

Once you submit the revision, we will assess your revised manuscript and your response to the reviewers' comments with our Academic Editor aiming to avoid further rounds of peer-review, although might need to consult with the reviewers, depending on the nature of the revisions.

**IMPORTANT - SUBMITTING YOUR REVISION**

3. Resubmission Checklist

a) *PLOS Data Policy*

- Supplementary files (e.g., excel). Please ensure that all data files are uploaded as 'Supporting Information' and are invariably referred to (in the manuscript, figure legends, and the Description field when uploading your files) using the following format verbatim: S1 Data, S2 Data, etc. Multiple panels of a single or even several figures can be included as multiple sheets in one excel file that is saved using exactly the following convention: S1_Data.xlsx (using an underscore).

- Deposition in a publicly available repository. Please also provide the accession code or a reviewer link so that we may view your data before publication.

Fig. 1B, C; Fig. 2E; Fig. 4I; Fig. 5A-D; Fig. 6A and Fig. S1A-E

***Please also note that for figures containing FACS data, we ask that you provide FCS files along with the picture showing the successive plots and gates that were applied to the FCS files to generate the figure.

b) *Blot and Gel Data Policy*

c) *Published Peer Review*

Sincerely,

Ines

--

Ines Alvarez-Garcia, PhD

Senior Editor

PLOS Biology

Reviewers' comments

Rev. 1:

The study done by Yuan et al. reports molecular analysis of the interaction between Bak and VDAC2, which constitute a key complex of the mitochondria-dependent apoptosis. Despite its significance in apoptosis regulation, this complex has been less investigated at an atomic level compared to the Bcl-2 family protein complexes, presumably because of technical difficulty. In this study, the authors attempted to address the molecular aspect of the Bak-VDAC2 intermolecular interaction, which were performed by a combination of deep scanning mutagenesis coupled with various biochemical approaches, including structural modeling, and cellular analyses. I consider that these results are interesting and scientifically important, and should substantially contribute to the precise understanding of Bak-mediated apoptosis. However, several points should be addressed before final decision, which are listed below.

>Page 5

- The authors described that the residues at the Bak hydrophobic groove, such as Arg88 and Tyr89, are proximal to the VDAC2 β10-11 loop, which was verified by cysteine cross-linking analysis. Figure 2B shows the location of those residues, but the hydrophobic groove is undiscerned at the structure of Bak. I suggest the authors to show it more clearly. Maybe surface representation of the Bak structure will be helpful.

>Page 6

- The authors also mentioned that the canonical hydrophobic groove of Bak is involved in its interaction with VDAC2. This groove is well known as the binding pocket of the BH3 domain from the Bcl-2 family proteins, implying the possibility of the β10-11 loop VDAC2 and the Bak-binding BH3 domains are competitive to associate with Bak. Therefore, I considered that the structural comparison analysis of Bak in a VDAC2-bound (modeled in this study; shown in Figure 3) and in BH3-bound forms (determined by numerous previous studies) should be added to this study.

>Page 8-9

- It is written that stabilisation of the BAK-VDAC2 complex interferes with apoptosis. But is it certain that the cells indeed underwent "apoptotic" death? The cell death analysis should be compensated by additional methods to confirm it, such as the caspase cleavage assay.

> Page 10

It is written that "we investigated cytochrome c release and cell death" with a delete line that should be removed.

> Page 10

The effects of WEHI-9625 were tested in MEFs expressing wild type VDAC2 or VDAC2A172W, but not in cells expressing VDAC2A172L. Is there any reason?

Rev. 2: Tudor Moldoveanu – note that this reviewer has signed his review

This timely, well-executed, and well-presented study, based on an initial, elegant deep scanning mutagenesis experiment, reveals additional insights into the direct interactions between BAK (and BAX) and VDAC2, which regulate apoptotic response. This interaction could be targeted to inhibit (or activate) apoptosis as suggested by a mouse BAK selective compound WEHI-9625 which stabilizes the BAK-VDAC2 inhibitory complex, although the mechanism of this inhibitor is yet to be defined in future structural investigations as the authors suggest.

I have one question related to the use of MCL-1 and BCL-xL selective inhibitors to trigger apoptosis. Have the authors also used a more potent combination of S63845 and ABT-263 or ABT-737 to induce apoptosis and under those conditions is the VDAC2-mediated inhibition of BAK/BAX-induced apoptosis more readily alleviated between the differentVDAC2 A172 mutants? This information would be very informative for targeting the VDAC2/BAK/BAX inhibitory axis.

A couple of typos include:

Page 9 A113852 should probably by A1331852.

Page 10 The strikethrough should be deleted.

Rev. 3: Christoph Borner – note that this reviewer has signed his review

This is an impressive, nicely performed and convincing study further determining the interaction between VDAC2 and BAK and its consequence for BAK inhibition and activation. The authors show that A172 within the VDAC2 β10-11 loop interacts with the hydrophobic binding pocket of BAK. Obstructive labelling disrupts this interaction, A172 can be crosslinked to the BAK pocket, and mutations of A172 affect BAK localization, the stability of the VDAC2-BAK complex and the proapoptotic activity of BAK. It seems that the β10-11 loop structure of VDAC2 directly interacts with the BAK pocket rather than a helix structure as it is characteristic for the classical BH3-peptide/BCL-2 interaction. But as for Bcl-2 family interactions, BH3 proteins also seem to compete with VDAC2 for the BAK groove, thereby releasing BAK for oligomerization, cytochrome c release and apoptosis.

A major part of the study shows that the type of substitution made to VDAC2 A172 can have variable impact on the stability of the VDAC2-BAK complex. Mutations to medium sized hydrophobic sidechains like Ile, Leu, Cys stabilise the complex, whilst the mutation to a large hydrophobic residue like Trp is less able to restrain BAK. Interestingly, a previously developed compound WEHI-9625 seems to phenocopy the stabilizing effect of Leu or Ile mutations at A172 thus explaining why WEHI-9625 is an effective BAK inhibitor.

Since VDAC2 is also required for BAX localization to mitochondria and regulating its pro-apoptotic activity, the authors also studied the effect of the A172 mutants on BAX. As shown in Figure 4F, these mutants had opposite effects on BAX interaction. A172L/I/C VDAC2 mutations destabilized the VDAC2-BAX interaction while more BAX becomes associated with VDAC when A172 is mutated to Trp. However, in Bak-/-Vdac2-/- cells the VDAC2 A172L/I mutations still had an inhibitory effect on BAX leading to diminished apoptosis in response to BH3-mimetic treatment (Figure 5C). The authors propose that while BAK is less freed from VDAC2 A172L/I/C, BAX becomes more cytosolic, thereby exerting the same effect, namely less apoptosis induction.

The VDAC2-BAK interaction studies are convincing and provide sufficient novelty for a publication in PLOS Biology. However, the interpretation of the VDAC2-BAX data is less clear.

Minor point:

What happens with the BAX pool that binds less to VDAC2 L172L/I/C in Figure 4F? In contrast to what the authors suggest, it most likely remains on mitochondria rather than being retrotranslocated to the cytosol (no increase in cytosolic BAX in Figure 4G). But in what form is BAX present on mitochondria under these circumstances? One possibility is that it is responsible for the minor tBID-triggered cytochrome c release from Vdac2-/- mitochondria (Figure 4H) or the BH3-mimetic-triggered cell death in Bak-/-Vdac2-/- MEFs expressing the VDAC2 L172L/I/C mutant (Figure 5C). In fact, diminished BH3-mimetic cell death due to VDAC2 L172L/I/C expression is more pronounced in Bax-/-Vdac2-/- (Figure 5A) than Bak-/-Vdac2-/- MEFs (Figure 5C). Why is this the case if the impact of the VDAC2 L172L/I/C mutants on BAX and BAK are the same?

Can the authors show that in Vdac2-/- cells expressing the VDAC2 L172L/I/C mutants (Figure 4F), the BAX released from these mutant proteins oligomerizes (Blue native, crosslinking), especially upon treatment with BH3 mimetics (on cells or mitochondria in vitro)? Treating these cells or their isolated mitochondria with BH3 mimetics would not imply BAK because it is blocked by the strong VDAC2 L172L/I/C interaction and involve only BAX. I think this is an important experiment given the proposition of the authors to treat diseases caused by mitochondrial apoptosis with WEHI-9625. If WEHI-9625 only stabilizes the VDAC-2/BAK interaction and does not effectively retrotranslocate BAX to the cytosol, then BAX-mediated MOMP may still occur and the drug would not be as efficient as thought.

Rev. 4:

In this study, Yuan and colleagues extensively define the functional relationship between VDAC2 and BAK (with some latter focus on BAX). Using the knowledge that disruption of VDAC2/BAK binding leads to degradation of BAK, they nicely couple this with deep-mutational scanning to identify putative VDAC2/BAK interacting residues. They then progress to validate and define these residues through cysteine and obstructive labelling of VDAC2, later defining a likely interaction between VDAC2 and BAK occurs via the canonical BAK hydrophobic groove. Finally, they demonstrate functional relevance of the VDAC2/BAK interacting residues, showing that stabilization of the interaction inhibits MOMP/cell death, whereas destabilization promotes BAK induced death.

In my

---

## [Editor Report · Decision Letter 2]

5 Apr 2024

Dear Dr Dewson,

Thank you for the submission of your revised Research Article entitled "Key residues in the VDAC2-BAK complex can be targeted to modulate apoptosis" for publication in PLOS Biology. On behalf of my colleagues and the Academic Editor, Pascal Meier, I am delighted to let you know that we can in principle accept your manuscript for publication, provided you address any remaining formatting and reporting issues. These will be detailed in an email you should receive within 2-3 business days from our colleagues in the journal operations team; no action is required from you until then. Please note that we will not be able to formally accept your manuscript and schedule it for publication until you have completed any requested changes.

PRESS

Sincerely, 

Ines

--

Ines Alvarez-Garcia, PhD

Senior Editor

PLOS Biology
